# Discerning Selfiers: Differences between Taking and Sharing Selfies

**DOI:** 10.3390/bs14080732

**Published:** 2024-08-22

**Authors:** Charisse L’Pree Corsbie-Massay, Rikki Sargent McLaren

**Affiliations:** 1Department of Communications, S. I. Newhouse School of Public Communications, Syracuse University, Syracuse, NY 13244, USA; 2Department of Psychology, College of Arts and Sciences, Syracuse University, Syracuse, NY 13244, USA

**Keywords:** selfie, self-esteem, control, well-being, social media, sharing

## Abstract

Selfies provide unique opportunities to explore, document, and interact with the self through photography. However, the inherent intrapersonal affordance of self-portraiture becomes a unique manifestation of masspersonal theory when selfies are shared, a step that is often assumed but not unpacked in selfie research. Therefore, it is essential to understand when and for whom these intrapersonal and interpersonal moments evolve into masspersonal moments or communication episodes. This study uses a novel strategy to investigate selfie behavior–the likelihood of taking and sharing selfies–to assess individual differences between selfie-takers, or “selfiers”. Discerning selfiers–people more likely to take than share selfies–report greater control and self-esteem compared to non-discerning selfiers, who are equally as likely to take as share selfies. Furthermore, control mediates the effect of selfier type on self-esteem; discerning selfiers report that others are *not* in control of their life, resulting in greater security regarding others’ perceptions of them. The current findings reveal the unique effects of differential deployment of selfies.

## 1. Introduction

Selfies, a popular form of self-representation in the 21st century, are pictures of oneself taken with a smartphone or digital device. This unique form of digital self-portraiture provides opportunities to explore, document, and interact with the self and others. People who take selfies are motivated by self-approval and belonging [1,2], impression management [3], attention seeking, communication, and entertainment [4], as well as documentation or archiving [1,4,5]. Although several studies have explored selection and editing behaviors [6,7], only a few studies have distinguished between taking and sharing selfies [8,9,10], and sharing selfies is considered to be inherent to, or definitive of, selfie use. Disregarding the importance of deliberation in making personal moments available to a mass audience limits our understanding of the unique processes, drives, and relationships underlying self-documentation and self-representation via selfies, especially given that selfies are powerful opportunities for self-reflection, indicating both an opportunity for “mass communication and interpersonal communication simultaneously” [11].

This research connects two important bodies of work: qualitative or content analysis work assessing different selfie genres, and traditional quantitative psychological surveys investigating motivation and individual differences. Using a novel approach that assesses differences in taking and sharing across different scenarios, the current study poses the following questions: (1) Is taking selfies (i.e., self-documentation) a distinctly different behavior from sharing selfies (i.e., self-representation)? And (2) do people who share selfies deliberately exhibit different patterns between constructs? Understanding different types of selfiers—that is, who, when, why, and how people selfie—is essential to understanding the motivations and outcomes of this twenty-first century phenomenon. Although the term “selfier” is not commonly used in research concerning selfies, we believe that this is an important means of shifting the discourse to focus on the uniqueness of individual practices.

### 1.1. Studying Selfies

Approximately two-thirds of Americans have taken and shared a selfie online, yet the most popular words associated with selfies are “narcissistic” and “annoying” [12]. Despite this prevalence, research and public discourse regularly frames selfies as having negative psychological effects (e.g., [13,14]): selfie posting is correlated with greater narcissism (e.g., [15,16,17,18]), histrionic personality [19,20], the development of the selfitis scale [21], other addiction behaviors [22], and decreased self-esteem and body positivity (e.g., [23,24,25,26]). This work often implicitly and explicitly posits selfies as the “problem” or “issue” to be solved, thereby framing self-esteem as an outcome of extensive selfie-engagement [27].

However, few studies consider control as a mediating and moderating factor, and experimental work, which many see as a standard in social sciences, often requires users to take selfies, regardless of preference or context, effectively stripping the control that comes from the personal choice to take and share selfies. Recent studies have found that deliberative selfie posting, or posting selfies that one believes will have the most effective impact, may be associated with positive impacts [7,28], and that editing or manipulating the representations of oneself is associated with negative outcomes [26,29]. Furthermore, selfie-making is also associated with positive intrapersonal outcomes including creativity [30], self-promotion, and self-confidence [22,26,31], and goal-directed selfies (e.g., taking a smiling selfie) result in greater positive affect [32]. Qualitative studies reveal that selfies communicate one’s thoughts and feelings when posted to social media [33], and selfie activism exemplifies the opportunity to take control of one’s own image and affect society at large [34,35,36]. The opportunity to cultivate control via selfies may underlie these positive outcomes.

Underlying these disparities in findings is a lack of consensus regarding the operationalization of selfie-related measures. In many cases, “selfie behavior” or “selfie engagement” is operationalized as a singular construct that assesses selfie posting [37,38,39]. This is unsurprising given that multiple dictionary entries require selfies to be uploaded to social media, but several studies have shown that people take and save more photos than they share [40,41]. This gap in the literature is glaring given the role of photography as documentation and interpersonal communication [42]. Because few studies distinguish between selfies taken and selfies shared, the potential benefits of selfies “taken for oneself” [43] and the unique experiences of deliberative posting have only recently been investigated [7]. A notable exception to this oversight is the Selfie Stadium Model proposed by de Vaate and colleagues [44], which unpacks the extended choice process that occurs from before one decides to take a selfie to ultimately posting it on social media. This model incorporates the “imagined audience” [45,46] or “imaginary” audience [47], the motivations for “selfie-making” alongside the social expectations of selfie-making (i.e., preoccupation), and the selection and editing process that occurs after a series of selfies have been snapped before finally posting.

### 1.2. Personal and Masspersonal Photography

Photographs for private citizens (i.e., not public figures) have historically been a space of interpersonal and intrapersonal communication: taking a picture of oneself has traditionally been something that was shared with other people in one’s inner circle due to the inherent nature of the technology [42]. With advances in digital technology and social media platforms, content originally produced as a means of interpersonal and intrapersonal communication can now become mass communications, and influencers and other digital entrepreneurs parlay this expectation of interpersonal communications into a career in mass media: an interpersonal sentiment becomes accessible to a mass audience.

Considering the masspersonal potential of selfies reframes important demographic differences in selfie-behavior. Women post more selfies than men [20,48,49], adolescents and younger adults post more selfies than older adults [19,49], and non-exclusively heterosexual people post more selfies than heterosexual people [19]. Similarly, qualitative research showcases how marginalized racial and ethnic groups use selfies to share their own stories and experiences without the gatekeeping of traditional media institutions [50,51,52,53]. Selfies are a form of identity affirmation [54] and, through this lens, discriminated groups, including women, younger people, and LGBTQ+ individuals use selfies to affirm and amplify their own representation [53,55,56,57,58]. Marginalized individuals use selfies as a means of attaining control through digital technologies to combat systematic denial of control in society [59,60] and understanding the patterns in taking and sharing of selfies can help illuminate how this practice connects to reclaiming a sense of control.

This exploratory study begins to connect these interdisciplinary dots by considering differences between taking and sharing selfies across different scenarios to better understand the practices of different selfiers. Although we acknowledge that one must take a selfie before one can share a selfie, we posit that taking and sharing are two related but distinct behaviors (H1), i.e., whereas taking is motivated by self-documentation, sharing is motivated by self-representation [2], and different patterns of taking and sharing will emerge in different types of selfiers (H2).

Further, we explore the correlation between selfie behaviors and control by selfier type to understand the agentic benefits of the selfie-making process. Given the prior research regarding deliberative selfie-making and selfie practices among marginalized populations, we hypothesize that selfiers who report greater likelihood of taking than sharing selfies will report more positive psychological well-being (e.g., less reliance on external feedback, greater self-esteem, H3) through greater reported control (H3a).

## 2. Materials and Methods

### 2.1. Sample

Participants were required to be 18 years of age or older, fluent in English, and have an account on at least one social networking site. Of the 216 participants who completed the survey, 22 were excluded because they were not at all likely to take a selfie in any of the featured scenarios (see below), three were excluded as age outliers because they were more than three standard deviations greater than the mean (i.e., over 45 years of age), and ten were excluded for missing data, resulting in a final analytic sample of *N* = 181 (age range 18–45 years; *M_AGE_* = 30.79; *SD_AGE_* = 6.00; 37.0% female; 77.3% born in the United States; 23.2% Asian; 10.5% Black or African American; 59.7% White; 4.4% Hispanic or Latino; 1.7% Multiracial/Mixed; 0.6% American Indian or Alaskan Native). Participants reported having an average of 5.34 (*SD* = 2.33) social networking accounts.

### 2.2. Procedure

Participants were recruited through Amazon’s Mechanical Turk and received $1 in compensation. Upon reading the recruitment materials, participants clicked a link that took them to a Qualtrics survey, where they read an informed consent sheet and, if they consented to participate in the study, responded to a series of demographic questions including their general online behaviors (e.g., checking social media, posting status updates) and preferred social networking sites. Participants then responded to a series of individual difference measures and an experimenter-generated selfie likelihood instrument (described below) before being debriefed and thanked.

### 2.3. Measures

In the interest of space, only the measures featured in the current analysis are described in this section. All measures and materials are available on OSF (https://osf.io/hgs9z/ accessed on 18 August 2024). Zero-order correlations among the primary outcome variables described below (i.e., perceptions of feedback, self-esteem, and locus of control) are additionally available on OSF.

The perceptions of feedback scale [49] was originally designed to assess students’ perceptions of, and preferences for, feedback. The original 15-item scale was adapted by prompting participants to consider their general reactions to online feedback (e.g., likes, comments, shares) from their extended social network in response to their content posted online. Featured items included “When I receive a lot of online feedback, I feel encouraged” and “Online feedback tells me what the expectations of other people are” (Alpha = 0.862).

The state self-esteem measure [50] assesses one’s self-esteem at a given point in time on three dimensions via 20 items: performance or being secure in one’s abilities (e.g., “I feel confident about my abilities”; Alpha = 0.856), social or feelings of security as to how one is perceived by others (e.g., “I am worried about what other people think of me”; Alpha = 0.906), and appearance or a positive sense of one’s presentation to others (e.g., “I am pleased with my appearance right now”; Alpha = 0.813). Performance and social were significantly correlated (*r* = 0.633, *p* < 0.001), performance and appearance were significantly correlated (*r* = 0.359, *p* < 0.001), but appearance and social were not correlated (*p* = 0.126). State self-esteem recognizes the association between participants’ current state and their anticipated selfie-intentions, and avoids claims of causation.

Locus of control [51], a 9-item measure, assesses the extent to which an individual feels that they are in control of their own life (i.e., internal control, e.g., “My life is determined by my own actions”; Alpha = 0.783), or whether it is controlled by external factors like others (i.e., external control, e.g., “My life is chiefly controlled by powerful others”; Alpha = 0.894) or chance (e.g., “To a great extent, my life is controlled by accidental happenings”; Alpha = 0.873). Internal and external were correlated (*r* = −0.222, *p* = 0.018), internal and chance were not correlated (*p* = 0.160), and external and chance were correlated (*r* = 0.810, *p* < 0.001).

Participants then completed a researcher-generated scale to assess intentions to take and share selfies across fifteen different scenarios using a five-item Likert-style bipolar scale (1 = I am NOT AT ALL likely; 5 = I am VERY likely; capitalization was purposeful to ensure that participants were aware of the response options given that the instrument was novel). Each scenario was presented separately and both items (i.e., likelihood to take, likelihood to share) were displayed simultaneously, allowing participants to consider the constructs alongside each other. The list of scenarios was developed through a close review of selfie types that were discussed online at the time of data collection (e.g., gym selfies, couple selfies, special occasion selfies); although not exhaustive, the scenarios were varied and allowed for a robust consideration of selfies in different contexts and emotions. Items are available in Table 1.

### 2.4. Data Analysis

Data were analyzed using SPSS version 28.0. We first assessed the descriptive statistics of the likelihood of taking and sharing selfie items. We then conducted a series of paired sample *t*-tests as a preliminary assessment of within-person differences concerning likelihood of taking vs. sharing selfies.

The likelihood of taking and the likelihood of sharing a selfie across fifteen different scenarios were then subjected to an exploratory factor analysis (EFA) with varimax rotation to clarify the relationships between factors. The rotated scree plot confirmed the distribution of factors before subjecting the subsequent scenario-based composites to a series of paired sample *t*-tests to determine differences between the likelihood of taking vs. sharing selfies in different situations.

Following this, we aimed to further differentiate taking and sharing behavioral patterns using k-means cluster analysis and hierarchal cluster analysis. Both the kml and hierarchal cluster functions were used on the scenario-based composites generated by the EFA with the default options, and patterns using two, three, and four clusters were examined. The final number of clusters was based on the fit criteria for each cluster pattern, and these clusters were then used in the following analyses.

A series of one-way Analysis of Variance (ANOVAs) and Multivariate Analyses of Variance (MANOVAs) were conducted to assess significant differences between clusters on the dependent variables; post-hoc ANOVAs and *t*-tests also confirmed differences. Finally, multivariate regression models were deployed using Hayes’ PROCESS macro program for SPSS to assess the predictive value of the selfie behavior clusters on individual differences and the mediating role of locus of control.

## 3. Results

Table 1 contains the descriptive statistics and paired sample *t*-test values for the likelihood of taking and the likelihood of sharing selfies across all 15 scenarios, as well as the factor loadings.

### 3.1. Item Analysis

A series of paired sample *t*-tests revealed significant differences between taking and sharing selfies for individual scenarios and for the composites. Participants were consistently more likely to take selfies compared to sharing selfies, and the likelihood of taking and sharing were significantly different for all but four scenarios (see Table 1).

### 3.2. Exploratory Factor Analysis

The Kaiser–Meyer–Olkin and Barlett’s tests on the 30 items showed a coefficient of 0.85 and χ^2^ = 5433.68 (*df* = 435, *p* < 0.001), indicating that a factor analysis was acceptable. An exploratory factor analysis was then performed on the 30 likelihood items to discover differences between scenarios. The varimax rotated matrix was extracted by principal axis factoring analysis, resulting in a six-factor solution. Absolute value of factor loadings below 0.40 were deleted. A review of the scree plot revealed three factors accounting for 47.70% of the variance. One scenario, silly outfit (i.e., “You put on an outfit and you feel that it looks silly”), loaded onto the second factor as well as its own factor—this scenario was aggregated into the second factor. Three scenarios, themed selfies, drinking with friends, and meeting a famous person each loaded onto their own factor and were dropped from further analyses, resulting in two factors accounting for 38.77% of the variance: pleasant moments (eight scenarios) and challenging moments (four scenarios).

When separating between taking and sharing selfies, the four dimensions demonstrated good reliability estimates: likelihood of taking selfies during pleasant scenarios (Alpha = 0.913), likelihood of sharing selfies from pleasant scenarios (Alpha = 0.914), likelihood of taking selfies during challenging scenarios (Alpha = 0.784), and likelihood of sharing selfies from challenging scenarios (Alpha = 0.836). Therefore, taking and sharing composites for pleasant and challenging scenarios were calculated and deployed in the subsequent analyses. Paired sample *t*-tests conducted on the composites revealed a significant difference in pleasant scenarios—participants were more likely to take selfies (*M* = 3.12, *SD* = 1.11) than share selfies (*M* = 2.94, *SD* = 1.12), *t* (180) = 4.51, *p* < 0.001—and a marginal difference in challenging scenarios—participants were marginally more likely to take selfies (*M* = 1.94, *SD* = 0.99) than share selfies (*M* = 1.90, *SD* = 1.02), *t* (180) = 1.74, *p* = 0.084. Therefore, H1 was supported: selfie taking and selfie sharing are two related but distinct behaviors.

### 3.3. Cluster Analysis

Next, we aimed to differentiate between patterns of taking and sharing behaviors across scenarios. The kml method was computed for two-, three-, and four-cluster solutions. The three-cluster solution achieved convergence in six iterations, whereas the two- and four-cluster solutions converged in seven iterations.

The three-cluster solution is presented in Table 2. The first selfie behavior cluster is “discerning” (*n* = 63; 35%) and is characterized by the high likelihood of taking and sharing selfies in pleasant scenarios, but a low likelihood of taking and sharing selfies in challenging scenarios (i.e., these selfiers differ in their selfie behavior across scenarios). Discerning selfiers also reported significant differences between their likelihood of taking and sharing in pleasant scenarios and challenging scenarios, as well as significant differences between pleasant and challenging scenarios in their likelihood of taking and sharing. The second selfie behavior cluster is “non-discerning” (*n* = 44; 24%) and is characterized by a high likelihood of taking and sharing in pleasant and challenging scenarios (i.e., these selfiers take and share selfies regardless of the scenario). Non-discerning selfiers also reported no significant difference between likelihood of taking and sharing in either scenario composite, nor did they report differences between composites on the likelihood of taking or sharing. The third selfie behavior cluster is “low” (*n* = 74; 41%) and is characterized by a low likelihood of taking and sharing selfies in pleasant and challenging scenarios (i.e., these selfiers are less likely to take and share selfies regardless of the scenario). Even so, low selfiers exhibit differences between their likelihood of taking and sharing in pleasant scenarios and challenging scenarios, and they differentiate between pleasant and challenging scenarios in their likelihood of taking and sharing. Therefore H2 was supported: different patterns of taking and sharing emerged in different types of selfies.

### 3.4. Individual Differences by Selfier Types

A one-way ANOVA revealed significant differences in age between clusters (*F* (2,178) = 12.66, *p* < 0.001). Tukey’s HSD Test for multiple comparisons revealed that low selfiers were significantly older (*M* = 33.08, *SD* = 6.87) compared to discerning (*M* = 30.18, *SD* = 4.58; *p* = 0.009, 95% C.I. = [0.62, 5.19]) or non-discerning selfiers (*M* = 27.80, *SD* = 4.64, *p* < 0.001, 95% C.I. = [2.75, 7.83]), and discerning and non-discerning selfiers were not significantly different (*p* = 0.084). Compared to non-discerning selfiers, discerning selfiers are more likely to be women according to a chi-squared test of independence (73.2% of women; 50.0% of men; *χ*^2^ = 5.608, *p* = 0.014). There was no difference by race (i.e., white vs. not white) or birth country (i.e., USA vs. not USA; *p*s > 0.300).

A series of one-way ANOVAs and MANOVAs were conducted to quantify the psychological differences between selfie behavioral types. There was a significant effect of selfier type on perceptions of (online) feedback, state self-esteem, and locus of control. There was no effect on public self-consciousness or need to belong. Means, standard deviations, and *t*-tests comparing discerning selfiers with non-discerning and low selfiers are available in Table 3.

Perceptions of (online) feedback. There was a main effect of selfier type on perceptions of feedback (*F* (2,178) = 8.35, *p* < 0.001). Low selfiers reported significantly lower importance of online feedback to their self-worth compared to discerning and non-discerning selfiers, but there was no significant difference between discerning and non-discerning selfiers.

State self-esteem. There was a main effect of selfier type on all three subscales (*F* (6,352) = 15.63, *p* < 0.001, *η*^2^ = 0.210) and between-subjects effects revealed a main effect on each subscale, including performance (being secure in one’s abilities), social (feelings of security as to how one is perceived by others), and appearance (a positive sense as to how one presents to others). Non-discerning selfiers reported significantly lower performance and social state self-esteem compared to discerning and low selfiers, but there was no significant difference between low and discerning selfiers on performance or social self-esteem. Appearance self-esteem demonstrated a different pattern of results: low selfiers reported significantly lower appearance state self-esteem compared to both discerning and non-discerning selfiers, and there was no difference between non-discerning and discerning selfiers.

Therefore H3 was partially supported: although there was no difference between types of selfier regarding perceptions of feedback or appearance self-esteem, there was a significant difference between discerning and non-discerning selfiers on performance and social state self-esteem.

Locus of control. There was a main effect of selfier type on all three subscales (*F* (6,352) = 6.41, *p* < 0.001, *η*^2^ = 0.100), and between-subjects effects revealed a main effect on each subscale, including internal (one is in control of their life), external (others are in control of one’s life), and chance (one’s life is up to chance). According to Tukey post-hoc analyses, non-discerning selfiers reported significantly lower internal locus of control compared to discerning selfiers but were not significantly different from low selfiers; there was no significant difference between discerning and low selfiers. Similarly, non-discerning selfiers reported significantly greater beliefs that others controlled their lives and that their lives were controlled by chance compared to discerning or low selfiers. Again, there was no significant difference between discerning and low selfiers.

### 3.5. Mediation Analysis

The PROCESS macro program was used to perform two joint mediation analyses. The two analyses aimed to understand the mediating role of locus of control (internal, others, and chance) on the relationship between selfier type (dummy coded; discerning = 0, non-discerning = 1) and the two state self-esteem subscales that exhibited an effect of selfier type: security in one’s abilities (i.e., performance; analysis 1) and security in how one is perceived by others (i.e., social; analysis 2). All model pathway results are displayed in Table 4 and Figure 1.

The three locus of control variables did not mediate the relationship between selfier type and performance esteem (total indirect effect = 0.1779; 95% bootstrapped CI [0.1153 to 0.2535]; Table 4 (a), Figure 1a). However, the three locus of control variables mediated the relationship between selfier type and social esteem (total indirect effect = 0.2531; 95% bootstrapped CI [0.2860 to 0.7312]; Table 4 (b), Figure 1b).

For clarity, we conducted a third mediation analysis to assess the independent mediating effect of external control on the relationship between selfier type and social esteem. External control mediated the relationship between selfier type and social esteem, with a total indirect effect of 0.1935 (95% bootstrapped CI [0.0825to 0.3399]; Table 4 (c), Figure 1c).

Therefore H3a was supported, the difference between discerning and non-discerning selfiers on social esteem, or their security in how others perceived them, was mediated by the extent to which control of their lives was determined by others.

## 4. Discussion

Selfies provide opportunities to engage with oneself, which is correlated with a greater sense of psychological well-being through increased control and self-exploration [34,61]. Selfies also serve as a means of mass communication by broadcasting a preferred sense of self to the public [33,60]. Although some have argued that selfies can have beneficial effects on well-being, the process by which this occurs, and how publicly displaying the self subsequently impacts the individual, has not yet been quantified.

The current study sought to investigate distinct behavioral components of the selfie process—taking and sharing—and whether individuals who demonstrate a distinction between taking and sharing report better psychological well-being using a novel method of selfie-assessment with a diverse sample of online users. Individuals who are more likely to take than they are to share selfies (i.e., those that are deliberate about their masspersonal sharing) reported greater control over their own lives, which in turn was associated with greater self-esteem. This work sets a critical foundation for future research that differentiates between different types of selfiers and the role of mass communications in the individual’s personal sense of self.

### 4.1. Taking vs. Sharing Selfies

Contrary to the dictionary definition of selfies, which states that selfies must be shared via social media, and the studies investigating selfies, which often feature selfie-posting as the primary frequency metric, participants reported a significantly greater likelihood of taking a selfie than sharing a selfie. Furthermore, participants engaged with selfies differently across scenarios. Participants were more likely to take and share selfies in pleasant scenarios (e.g., time with friends, family, significant other, celebratory events) compared to challenging scenarios (e.g., crying, bad hair day, at the gym). Participants also demonstrated a significant difference between taking and sharing in pleasant scenarios, but not in challenging scenarios, given the low likelihood of taking selfies during challenging moments.

This seemingly obvious difference—that people take but do not post selfies—is ignored by many quantitative studies that assess selfie frequency only via social media posting (e.g., [1,2,3,4,5,16,62]). This research is one of the first studies to assess selfie-taking as separate from, but related to, selfie-posting. In doing so, we also demonstrated that selfie behaviors differ between scenarios. These findings demonstrate that the act of taking selfies must be contextualized and explored when investigating the psychological impact of selfies, and that taking a selfie should not be immediately associated with self-presentation.

### 4.2. Clusters of Selfiers

The cluster analysis revealed three distinct clusters of selfie behavior patterns. Most participants were low selfiers who reported that they were “not at all” likely to take or share selfies across scenarios, and reported significant differences between their likelihood of taking and sharing selfies across pleasant and challenging situations. Among high selfiers, two clusters appeared: discerning selfiers and non-discerning selfiers. Discerning selfiers were more likely to take and share selfies in pleasant scenarios compared to challenging scenarios, and reported a significant difference between their likelihood of taking and sharing in both scenario composites. Alternatively, non-discerning selfiers reported no difference between the likelihood of taking and sharing between pleasant and challenging scenarios, nor did they report a significant difference between taking and sharing in either composite. Discerning selfiers also reported higher internal locus of control, lower external locus of control (i.e., others, chance), higher social self-esteem, and higher performance self-esteem compared to non-discerning selfiers, and greater control mediated the relationship between selfier type and social self-esteem among high selfiers.

Prior studies have correlated selfies with lower self-esteem via social media feedback (e.g., likes; [23,24,26]), but the current findings indicate that these trends may be driven by non-discerning selfiers. Again, these studies value the public sharing of selfies in isolation and disregard the possibility that selfies—specifically selfies for oneself—can enhance control. Some studies have explored the relationship between editing one’s selfies and traditional psychosocial metrics [7,17,29]), but this technical form of control introduced another layer to the process of selfies—perfecting one’s self-presentation. Investigating the unique patterns of taking and sharing selfies reveals how the act of taking a selfie independently of sharing is correlated with an internal sense of control, without considering the complexity of controlling one’s public appearance.

### 4.3. Limitations and Future Directions

The present study is one of the first studies to investigate users’ tendencies to take and share selfies in different scenarios as separate but related constructs. In doing so, we complicated the assessment of selfies and revealed how selfies can be related to psychological well-being, as well as to control and agency. Although the current findings offer important preliminary insights into the relationship between selfie use, psychology, and mass distribution of personal content, there are several limitations that should be considered.

First, the study deployed a novel instrument to assess selfie behavior: intentions to take and share selfies in different scenarios. Although the current findings provide insight into how different people approach selfies in different scenarios, the list is not exhaustive and future work should consider additional scenarios. Similarly, the selfie process is extensive and includes choosing to take, curating images, editing, and posting, as well as strategic posting and deletion (i.e., choosing what time to post and removing images that do not elicit significant engagement), and future work should explore how selfie behaviors at each of these steps correlates with cognitive processes. Second, the study sample is relatively small and additional data are required to confirm the factor analysis of selfie scenarios, as well as the cluster analysis. Similarly, although the current sample is diverse with respect to age, gender, and nationality, additional analyses should increase subsample sizes to allow for within-group assessments.

Third, deploying validated selfie addiction scales, as well as traditional scales assessing constructs popular in prior selfie research (e.g., narcissism, objectification, body image concerns, social media engagement), may provide greater understanding of additional psychological patterns between discerning and non-discerning selfiers. In the case of selfie addiction, non-discerning selfiers may be unable to moderate or control their selfie-taking and -sharing, a behavioral outcome associated with addiction and one that is amplified in a masspersonal ecosystem. Finally, if discerning selfie behavior is associated with better psychological well-being, then future studies should explore how to foster discerning selfie behavior among digital producers. Content is produced and distributed almost instantaneously in the twenty-first century, therefore stopping to reflect on content prior to distribution may have significant positive effects on psychological well-being.

## 5. Conclusions

This study investigated the relationships underlying individual differences, the likelihood of taking selfies, and the likelihood of sharing selfies, as well as why individuals might take and share selfies, providing a framework for future research on this topic. Furthermore, non-discerning selfiers, or those who engage in mass sharing of selfies with less deliberation, reported lower state self-esteem along with less internal and greater external locus of control. Our work revealed that researchers must consider nuanced selfier types when exploring the psychological effects of selfies and individual processes that manifest in the masspersonal ecosystem. Altogether, we effectively bridged the subfields of media, psychology, and communication to understand how and why individuals use technology and media as a form of self-documentation and self-representation.

## Figures and Tables

**Figure 1 behavsci-14-00732-f001:**
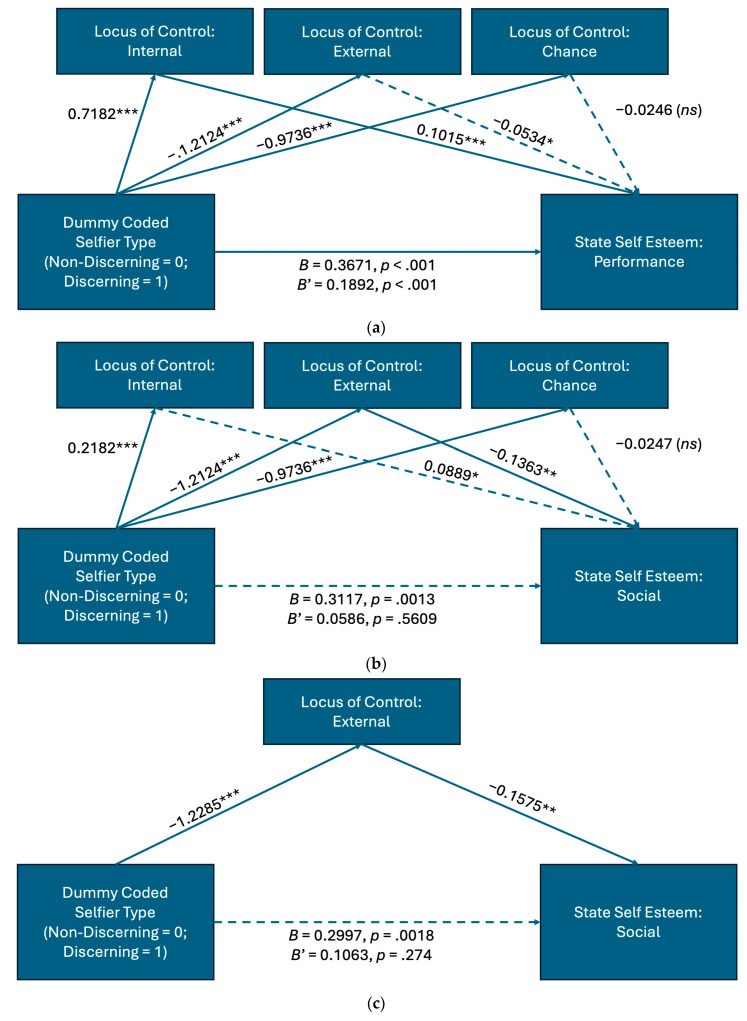
Mediation effects with unstandardized *B*s; *** *p* < 0.0001, ** *p* < 0.01, * *p* < 0.05; dashed lines indicate not significant at *p* < 0.01 level. (**a**): Predicting state self-esteem (SSE) performance by selfier type and three locus of control (LOC) subscales. (**b**): Predicting state self-esteem (SSE) social by dummy-coded selfier type and three locus of control (LOC) subscales. (**c**): Predicting state self-esteem (SSE) social by selfier type and locus of control (LOC) external. Note: *ns* indicates *p* > 0.05.

**Table 1 behavsci-14-00732-t001:** Item analysis: differences in the likelihood of taking and sharing selfies across 15 scenarios and factor loadings.

	Likelihood			Factor Loadings		
Scenario	Take M (SD)	Share M (SD)	Take vs. Share t (*p*-Value)	Factor	Take	Share
You are celebrating a regular personal event (e.g., birthday, anniversary).	3.16 (1.45)	3.01 (1.41)	2.24 (0.027)	Pleasant	0.77	0.64
You are spending time with your significant other.	3.02 (1.38)	2.77 (1.37)	3.46 (<0.001)	Pleasant	0.76	0.69
You are spending time with family.	2.83 (1.38)	2.71 (1.43)	1.93 (0.055)	Pleasant	0.73	0.69
You are spending time with friends.	3.19 (1.37)	3.00 (1.41)	2.93 (0.004)	Pleasant	0.61	0.55
You are attending a special event (e.g., graduation, concert).	3.22 (1.33)	3.09 (1.39)	1.99 (0.048)	Pleasant	0.58	0.56
You feel good about some part of your body, including your clothes, hair, etc.	2.87 (1.56)	2.69 (1.49)	3.15 (0.002)	Pleasant	0.55	0.54
You are on vacation.	3.69 (1.26)	3.48 (1.33)	3.53 (<0.001)	Pleasant	0.51	0.51
You are playing with your pet (e.g., cat, dog, hamster).	2.95 (1.47)	2.76 (1.50)	2.81 (0.005)	Pleasant	0.46	0.45
You are upset and have been crying.	1.59 (1.09)	1.58 (1.07)	0.33 (0.74)	Challenging	0.89	0.90
You are having a bad hair day.	1.74 (1.16)	1.69 (1.08)	1.34 (0.18)	Challenging	0.84	0.85
You put on an outfit and you feel that it looks silly.	2.28 (1.41)	2.19 (1.39)	1.75 (0.081)	Challenging	0.47	0.56
You are at the gym.	2.13 (1.38)	2.13 (1.40)	0.00 (1.00)	Challenging	0.46	0.62
You are in the presence of a famous person that you admire.	3.66 (1.29)	3.48 (1.38)	3.02 (0.003)	Single 1	0.85	0.74
You have been drinking with friends and you are having a good time.	2.84 (1.45)	2.54 (1.35)	4.19 (<0.001)	Single 2	0.57	0.68
Several of your friends have posted themed selfies and invited you to post a themed selfie as well.	3.19 (1.42)	3.00 (1.41)	3.45 (<0.001)	Single 3	0.74	0.84

Note. Paired-sample *t*-tests were used to identify differences between the take and share likelihoods for each scenario (*df* = 180). The last three scenarios loading onto their own factors and were dropped from further analysis.

**Table 2 behavsci-14-00732-t002:** Cluster analysis: differences in the likelihood of taking and sharing selfies within and between pleasant and challenging scenarios.

	Likelihood		
Selfier Type	Take M (SD)	Share M (SD)	Take vs. Share t (*p*-Value)
Discerning (n = 63)			
Pleasant Scenarios	4.07 (0.53)	3.89 (0.62)	3.05 (0.003)
Challenging Scenarios	1.78 (0.48)	1.67 (0.47)	3.74 (<0.001)
**Pleasant vs. Challenging t (*p*-value)**	32.08 (<0.001)	2.87 (<0.001)	
Non-discerning (n = 44)			
Pleasant Scenarios	3.53 (0.69)	3.46 (0.58)	0.85 (0.400)
Challenging Scenarios	3.38 (0.64)	3.44 (0.62)	0.79 (0.435)
**Pleasant vs. Challenging t (*p*-value)**	1.51 (0.138)	0.267 (0.790)	
Low (n = 74)			
Pleasant Scenarios	2.06 (0.69)	1.82 (0.58)	3.61 (<0.001)
Challenging Scenarios	1.21 (0.38)	1.17 (0.35)	2.71 (0.008)
**Pleasant vs. Challenging t (*p*-value)**	10.28 (<0.001)	9.61 (<0.001)	

Note. Paired sample *t*-tests were used to identify differences between the take and share likelihoods for each scenario composite (*df*_DIS_ = 62; *df*_ND_ = 43; *df*_LOW_ = 73). Paired sample *t*-tests were used to identify differences between the pleasant and challenging scenarios for the likelihood of taking and sharing selfies (*df*_DIS_ = 62; *df*_ND_ = 43; *df*_LOW_ = 73).

**Table 3 behavsci-14-00732-t003:** Individual differences analysis: individual differences by selfier type.

Individual Difference	Discerning M (SD)	Non-Discerning M (SD)	Low M (SD)	Discerning vs. Non-Discerning t (*p*-Value)	Discerning vs. Low t (*p*-Value)	Non-Discerning vs. Low t (*p*-Value)
Public Self-Consciousness	2.76 (0.72)	2.75 (0.52)	2.67 (0.80)	0.07 (0.947)	0.65 (0.514)	0.57 (0.569)
Need to Belong	3.25 (0.73)	3.11 (0.38)	2.98 (0.68)	1.14 (0.259)	2.13 (0.035)	1.10 (0.275)
Perceptions of (Online) Feedback	3.55 (0.53)	3.45 (0.48)	3.18 (0.60)	1.03 (0.306)	3.82 (<0.001)	2.52 (0.013)
SSE Performance	1.80 (0.27)	1.43 (0.20)	1.75 (0.26)	7.70 (<0.001)	1.05 (0.295)	6.99 (<0.001)
SSE Social	3.90 (0.87)	3.00 (0.94)	3.85 (0.97)	5.11 (<0.001)	0.40 (0.693)	4.62 (<0.001)
SSE Appearance	4.19 (0.72)	4.27 (0.61)	3.88 (0.70)	0.57 (0.570)	2.61 (0.010)	3.09 (0.003)
LOC Internal	4.90 (0.85)	4.27 (1.06)	4.65 (0.98)	3.39 (<0.001)	1.48 (0.142)	2.05 (0.042)
LOC External	2.68 (1.26)	3.97 (0.99)	2.96 (1.34)	5.63 (<0.001)	0.29 (0.770)	5.87 (<0.001)
LOC Chance	2.83 (1.17)	3.93 (1.13)	2.76 (1.30)	4.84 (<0.001)	0.30 (0.764)	4.91 (<0.001)

Note. Analysis of variance tests were used to identify differences between selfier type for each individual difference measure (*df* = 178). Paired sample *t*-tests were used to identify differences between discerning vs. non-discerning selfiers (*df* = 107), and between discerning vs. low selfiers (*df* = 135) for each individual difference measure. SSE = state self-esteem. LOC = locus of control. The performance state self-esteem subscale was square root-transformed.

**Table 4 behavsci-14-00732-t004:** Results of path analyses. (**a**): Predicting state self-esteem (SSE) performance by dummy-coded selfier type (non-discerning = 0, discerning = 1) and three locus of control (LOC) subscales. (**b**): Predicting state self-esteem (SSE) social by dummy-coded selfier type (non-discerning = 0, discerning = 1) and three locus of control (LOC) subscales. (**c**): Predicting state self-esteem (SSE) social by dummy-coded selfier type (non-discerning = 0, discerning = 1) and locus of control (LOC) external.

Path			St *ß*	Unst B	se	t	*p*	LLCI	ULCI
(**a**)									
Selfier Type	→	LOC Internal	0.7249	0.7182	0.1809	3.9714	<0.001	0.3598	1.0766
Selfier Type	→	LOC External	−0.8964	−1.2124	0.2373	−5.1087	<0.001	−1.6828	−0.7421
Selfier Type	→	LOC Chance	−0.7712	−0.9736	0.2282	−4.2656	<0.001	−1.4259	−0.5213
LOC Internal	→	SSE Perform	0.3372	0.1015	0.0197	5.1438	<0.001	0.0624	0.1406
LOC External	→	SSE Perform	−0.2423	−0.0534	0.0237	−2.2503	0.0265	−0.1005	−0.0064
LOC Chance	→	SSE Perform	−0.1753	−0.0414	0.0246	−1.6825	0.0954	−0.0902	0.0074
*Selfier Type*	*→*	*SSE Perform*	*1.234*	*0.3671*	*0.0465*	*7.8988*	*<0.001*	*0.2750*	*0.4592*
Selfier Type	→	SSE Perform’	0.6346	0.1892	0.0433	4.3654	<0.001	0.1033	0.2751
(**b**)									
Selfier Type	→	LOC Internal	0.7249	0.2182	0.1809	3.9714	<0.001	0.3598	1.0766
Selfier Type	→	LOC External	−0.8964	−1.2124	0.2373	−5.1087	<0.001	−1.6828	−0.7421
Selfier Type	→	LOC Chance	−0.7712	−0.9736	0.2282	−4.2656	<0.001	−1.4259	−0.5213
LOC Internal	→	SSE Social	0.1735	0.0889	0.0457	1.9442	0.0545	−0.0017	0.1795
LOC External	→	SSE Social	−0.3633	−0.1363	0.0550	−2.4786	0.0148	−0.2453	−0.0273
LOC Chance	→	SSE Social	−0.0613	−0.0247	0.0570	−0.4325	0.6663	−0.1377	0.0883
*Selfier Type*	*→*	*SSE Social*	*0.6141*	*0.3117*	*0.0945*	*3.2986*	*0.0013*	*0.1244*	*0.4989*
Selfier Type	→	SSE Social’	0.1154	0.0586	0.1004	0.5833	0.5609	−0.1405	0.2577
(**c**)									
Selfier Type	→	LOC External	−0.9077	−1.2285	0.2349	−5.2287	<0.001	−1.6940	−0.7629
LOC External	→	SSE Social	−0.4211	−0.1575	0.0350	−4.4974	<0.001	−0.2269	−0.0881
*Selfier Type*	*→*	*SSE Social*	*0.5921*	*0.2997*	*0.0939*	*3.1928*	*0.0018*	*0.1137*	*0.4858*
Selfier Type	→	SSE Social’	0.2100	0.1063	0.0968	1.0984	0.2744	−0.0855	0.2980

## Data Availability

All measures and materials are available on OSF (https://osf.io/hgs9z/ accessed on 18 August 2024).

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
