# Peer review of "Discerning Selfiers: Differences between Taking and Sharing Selfies"

_behavsci, 2024, doi:10.3390/bs14080732_

Round 1

Reviewer 1 Report

Comments and Suggestions for Authors

The presented work on selfie behavior is very valuable to the community. The manuscript is well written. Only the first paragraph of the introduction needs some work due to minor typing and spelling mistakes. Furthermore I would ask to specifiy the terms" selfier" early in the intro, as it is already used in text. The sentence in Line 26 Prior research... is hard to understand. It could also be considered to make the intro more concise. The rest of the mauscript is of excellent quality. 

Author Response

Thank you to the Editor and Reviewers at Behavioral Science for their consideration of our manuscript, “Discerning Selfiers: Differences between taking and sharing selfies” (behavsci-3082030). We have carefully reviewed the comments and have edited our manuscript accordingly. We believe that the current version is a marked improvement and we hope that we have addressed the reviewers’ concerns. 

Below, we have outlined our responses. The reviewer’s comments are in bold, our responses are in regular print, and insertions into the article are in italics and indented. 

Comment 1 The presented work on selfie behavior is very valuable to the community. The manuscript is well written. 

Thank you for this positive feedback!

Comment 2 Only the first paragraph of the introduction needs some work due to minor typing and spelling mistakes…  The sentence in Line 26 Prior research... is hard to understand. It could also be considered to make the intro more concise.

Thank you for your patience and careful reading. We have reviewed the entire manuscript for instances where our grammar could be improved. In the first paragraph specifically, we have streamlined and juxtaposed the findings regarding selfie motivations and selfie operationalization to better establish the value of our work (also mentioned by other reviewers). We have also removed the confusing sentences at line 26. The first paragraph now reads…

Selfies, a popular form of self-representation in the 21st century, are pictures of oneself taken with a smartphone or digital device. This unique form of digital self-portraiture provides opportunities to explore, document, and interact with the self and others. People who take selfies are motivated by self-approval and belonging [1,2], impression management [3], attention seeking, communication, and entertainment [4], as well as documentation or archiving [1,4,5]. Although several studies explore selection and editing behaviors [6,7], only a few studies distinguish between taking and sharing selfies [8–10] and sharing selfies is considered to be inherent to or definitive of selfie use. Disregarding the importance of deliberation in making personal moments available to a mass audience limits our understanding of the unique processes, drives, and relationships underlying self-documentation and self-representation via selfies especially given that selfies are powerful opportunities for self-reflection, indicating both an opportunity for “mass communication and interpersonal communication simultaneously” [11].  

Comment 3 I would ask to specify the terms" selfier" early in the intro, as it is already used in text. 

Thank you for this note. We have defined the term selfier the second paragraph of the introduction and explained why we have adopted this term…

From Section 1 Paragraph 2

Understanding different types of selfiers - that is, who, when, why, and how people selfie - is essential to understanding the motivations and outcomes of this twenty-first century phenomenon. Although the term “selfier” is not commonly used in the research regarding selfies, we believe that this is an important means of shifting the discourse to focus on the uniqueness of individual practices. 

Comment 4 The rest of the manuscript is of excellent quality. 

Thank You!

Reviewer 2 Report

Comments and Suggestions for Authors

The manuscript, “discerning selfiers” separates two distinct behaviors of selfie-taking and selfie-sharing and investigated their psychological implications. The study identifies two types of selfiers: discerning and non-discerning. The findings suggested that discerning selfiers take more selfies than they share and report higher levels of control and self-esteem. 

I appreciate the authors took a nuanced approach to study an important social media use behavior and exploring the connections with well-being.

Here are a few main concerns I have about the manuscript.

  1. Theoretical predictions 

The manuscript lacks well-developed research questions or hypotheses. The authors touch briefly about various motivations for selfie related social media behavior. However, how and why selfie-taking and selfie-sharing are connected to various motives and well-being are not fully developed in the manuscript. For example, self-esteem should be the cause of different selfie behavior, instead of an outcome. Similarly, there might be a main drive for both selfies taking and selfie posting.

  1. Related to the previous point, the manuscript fails to include recent research on selfies and social media use in general. For example, the selfies research summarized in the literature review on page 2 (line 59-69) was from 2015 to 2017. There were some more recent qualitative studies mentioned in a later literature review section, but they were not integrated with quantitative studies reviewed.  More importantly, the manuscript did not capture the recent research on social media and well-being, from new thinkings on self-presentation, self-disclosure, passive use and active use, to new large scale or review studies on the connections between social media use and well-being. 

  1. Selfie behavior. I am glad that the authors separate selfie taking to selfie posting. I do believe these are distinct behaviors. However, selfie taking and selfie posting are not the only selfie related behaviors. Selfie editing and selfie feedback monitoring are important behaviors as well. When the authors limit selfie behavior to only selfie taking and selfie posting, the study’s contribution is limited.  In addition, selfie behavior was measured with 15 different scenarios. Those 15 scenarios are common, but are they comprehensive? What’s the process the authors took to generate these 15 scenarios? What criteria did the authors use to make sure that these 15 scenarios are comprehensive? This is the main weakness of this research. The authors did not apply a solid method for the stimuli development. 

  1. There are a few smaller issues. For example, why did the study exclude participants born before 1970?

Comments on the Quality of English Language

The writing is clear. 

Author Response

Thank you to the Editor and Reviewers at Behavioral Science for their consideration of our manuscript, “Discerning Selfiers: Differences between taking and sharing selfies” (behavsci-3082030). We have carefully reviewed the comments and have edited our manuscript accordingly. We believe that the current version is a marked improvement and we hope that we have addressed the reviewers’ concerns. 

Below, we have outlined our responses. The reviewer’s comments are in bold, our responses are in regular print, and insertions into the article are in italics and indented. 

Comment 1 The manuscript, “discerning selfiers” separates two distinct behaviors of selfie-taking and selfie-sharing and investigated their psychological implications. The study identifies two types of selfiers: discerning and non-discerning. The findings suggested that discerning selfiers take more selfies than they share and report higher levels of control and self-esteem. I appreciate the authors took a nuanced approach to study an important social media use behavior and exploring the connections with well-being.

Response 1 Thank you for this summary and the positive feedback regarding our approach. 

Comment 2 The manuscript lacks well-developed research questions or hypotheses. 

Response 2 Thank you for this important gap in our writing. We have explicated that this study was exploratory in nature given that there is a lack of prior findings juxtaposing selfie taking and selfie posting. 

From 1.2 Paragraph 3 (Personal and Mass Personal Photography)

This exploratory study begins to connect these interdisciplinary dots by considering differences between taking and sharing selfies across different scenarios to better understand the practices of different selfiers. 

We have also amplified and enumerated our research questions and hypotheses for clarity. 

From 1.2 Paragraphs 3-4 (Personal and Masspersonal Photography)

…we posit that taking and sharing are two related but distinct behaviors (H1);  (i.e., whereas taking is motivated by self-documentation, sharing is motivated by self-representation [2]) and different patterns of taking and sharing will emerge in different types of selfiers (H2). 

Further, we explore the correlation between selfie behaviors and control by selfier type to understand agentic benefits of the selfie-making process. Given the prior research regarding deliberative selfie making and selfie practices among marginalized populations, we hypothesize that selfiers who report greater likelihood to take than share selfies will report more positive psychological well-being (e.g., less reliance on external feedback, greater self-esteem, H3) through greater reported control (H3a).  

Comment 3 The authors touch briefly about various motivations for selfie related social media behavior. However, how and why selfie-taking and selfie-sharing are connected to various motives and well-being are not fully developed in the manuscript. 

Response 3 Thank you for this point. We have explicated how taking and sharing are connected to different motivations, including a reference to a 2024 meta analysis by Felig & Goldenberg. 

Felig RN, Goldenberg JL. Selfie-evaluation: A meta-analysis of the relationship between selfie behaviors and self-evaluations. Pers Soc Psychol Bull. 2024;50(8):1227–50.

From 1.1 Paragraph 2 (Studying Selfies)

Recent studies have found that deliberative selfie posting - or posting selfies that one believes will have the most effective impact - may be associated with positive impacts [7,28], and that editing or manipulating the representations of oneself is associated with negative outcomes [26, 29].  Furthermore, selfie making is also associated with positive intrapersonal outcomes including creativity [30], self-promotion, and self-confidence [22, 26, 31], and goal-directed selfies (e.g., take a smiling selfie) result in greater positive affect [32]. 

Comment 4 For example, self-esteem should be the cause of different selfie behavior, instead of an outcome. Similarly, there might be a main drive for both selfies taking and selfie posting.

Response 4 We appreciate this important point. We agree that self-esteem should be considered as both a precursor and outcome of online behaviors in general, but recognize that much of the public discourse and research approaches frame self-esteem as an outcome of selfies.  

We have added literature regarding self esteem as both a predictor and an outcome of behaviors, and explicate how prior literature has framed selfies as having a negative effect on self-esteem and well being. In our earlier iteration, we describe the negative framing of selfies in the second paragraph of the introduction. We have moved this earlier to help highlight the flaws and gaps in earlier approaches.

From 1.1 Paragraph 1 (Studying Selfies)

Despite this prevalence, research and public discourse regularly frames selfies as having negative psychological effects (e.g., [13,14]): selfie posting is correlated with greater narcissism (e.g., [15–18]), histrionic personality [19, 20], the development of the selfitis scale [21] and other addiction behaviors [22], and decreased self-esteem and body positivity (e.g., [23–26]). This work often implicitly and explicitly posits selfies as the “problem” or “issue” to be solved, thereby framing self-esteem as an outcome of extensive selfie-engagement [27]. 

In addition, we opted to use state self-esteem in our methods as opposed to trait self-esteem in order to recognize the correlation between participants' current state and their responses to scenario-specific intentions. We have also reviewed our manuscript to emphasize the correlational nature of the findings and avoid any claims about causation.

From Section 2.3 Paragraph 3 (Measures)

We opted to use state self-esteem in our methods as opposed to trait self-esteem in order to recognize the correlation between participants' current state and their responses to scenario-specific intentions.

Comment 5 Related to the previous point, the manuscript fails to include recent research on selfies and social media use in general. For example, the selfies research summarized in the literature review on page 2 (line 59-69) was from 2015 to 2017. There were some more recent qualitative studies mentioned in a later literature review section, but they were not integrated with quantitative studies reviewed. 

Response 5 Thank you for pointing this out. This work is part of a longer historical study, but we eliminated the paragraph described by the reviewer and instead made our claims more explicit. We describe this earlier in this review in response to their comment regarding motivations, but we have re-pasted the paragraph here for clarity. 

Recent studies have found that deliberative selfie posting - or posting selfies that one believes will have the most effective impact - may be associated with positive impacts [7,28], and that editing or manipulating the representations of oneself is associated with negative outcomes [26, 29].  Furthermore, selfie making is also associated with positive intrapersonal outcomes including creativity [30], self-promotion, and self-confidence [22, 26, 31], and goal-directed selfies (e.g., take a smiling selfie) result in greater positive affect [32]. 

Comment 6 More importantly, the manuscript did not capture the recent research on social media and well-being, from new thinkings on self-presentation, self-disclosure, passive use and active use, to new large scale or review studies on the connections between social media use and well-being. 

Response 6 Thank you for this point. We hope that our edits as described above address the reviewer’s concern about framing the positive aspects of selfie-use. We opted not to frame the phenomenon of selfies through the lens of social media use in the interest of word count, but we acknowledge that this is emblematic of research gaps regarding social media use in general that is largely motivated by various social panic rhetoric. 

Comment 7 Selfie behavior. I am glad that the authors separate selfie taking to selfie posting. I do believe these are distinct behaviors. However, selfie taking and selfie posting are not the only selfie related behaviors. Selfie editing and selfie feedback monitoring are important behaviors as well. When the authors limit selfie behavior to only selfie taking and selfie posting, the study’s contribution is limited.  

Response 7 We recognize that there are different behaviors associated with the entire selfie process and have adjusted our language. We explicitly describe the research regarding selfie selecting and editing, and explicate the assumed inherent nature of selfie posting in selfie engagement.

From Section 1 Paragraph 1

Although several studies explore selection and editing behaviors [6,7], only a few studies distinguish between taking and sharing selfies [8–10] and sharing selfies is considered to be inherent to or definitive of selfie use.

We have also added this in our limitations as a call for future research to explore all of the nuanced decision points in the selfie-making and posting process.

From Section 4.3 Paragraph 2 (Limitations)

Similarly, the selfie process is extensive and includes choosing to take, curating images, editing, and posting, as well as strategic posting and deletion (i.e., choosing what time to post and removing images that do not elicit significant engagement), and future work should explore how selfie behaviors at each of these steps correlates with cognitive processes. 

Comment 8 In addition, selfie behavior was measured with 15 different scenarios. Those 15 scenarios are common, but are they comprehensive? What’s the process the authors took to generate these 15 scenarios? What criteria did the authors use to make sure that these 15 scenarios are comprehensive? This is the main weakness of this research. The authors did not apply a solid method for the stimuli development. 

Response 8 Thank you for this important point. We have elaborated the process by which we established these 15 scenarios and recognize that this is barely the tip of the iceberg. 

From Section 2.3 Paragraph 5 (Measures)

Participants then completed a researcher-generated scale to assess intentions to take and share selfies across fifteen different scenarios using a 5-item Likert-style bipolar scale (1 = I am NOT AT ALL likely; 5 = I am VERY likely; capitalization was purposeful to ensure that participants were aware of the response options given that the instrument was novel). Each scenario was presented separately and both items (i.e., likelihood to take, likelihood to share) were displayed simultaneously, allowing participants to consider the constructs alongside each other. The list of scenarios was developed through a close review of selfie types that were discussed online at the time of data collection (e.g., gym selfies, couple selfies, special occasion selfies); although not exhaustive, the scenarios were varied and allowed for a robust consideration of selfies in different contexts and emotions. Items are available in Table 1. 

From Section 4.3 Paragraph 2 (Limitations and Future Directions)

First, the study deploys a novel instrument in assessing selfie behavior: intentions to take and share selfies in different scenarios. Although the current findings provide insight into how different people approach selfies in different scenarios, the list is not exhaustive and future work should consider additional scenarios. 

Comment 9 Why did the study exclude participants born before 1970? 

Response 9 Thank you for this question. We did not exclude any participants from data collection, but three participants were age outliers (i.e., more than 3SD greater than the mean), and were excluded from data analysis. The final age range was 18-45. We hope that this clarification addresses the reviewer's concerns. 

From Section 2.1 Paragraph 1 (Sample)

Of the 216 participants who completed the survey, 22 were excluded because they were not at all likely to take a selfie in any of the featured scenarios (see below), three were excluded as age outliers because they were more than three standard deviations greater than the mean (i.e., over 45 years of age), and ten were excluded for missing data, resulting in a final analytic sample of N = 181…

Reviewer 3 Report

Comments and Suggestions for Authors

I thank the Editor for giving me this opportunity to go through this manuscript. I also applaud the author(s) for the efforts.

While the paper is interesting, the execution of the same needs improvement. Please see my comments as below:

·   The introduction of the paper is good but there lacks some clarity as to why it is important to conduct this research. Strong enough arguments are needed to justify the need of the research/study. Please note that without a clear introduction, your readers will struggle. They may feel confused when they start reading your paper. A good introduction should have Scope: The topic you’ll be covering; Context: The background of your topic and Importance: Why your research matters in the context of an industry or the world, to guide your reader from a general subject area to the narrow topic that your paper covers. Please neatly address the valid gap(s) and research objectives of the paper based on the strong motivation. Include research questions that address the gap(s) identified.

      Please note that theoretical grounding and the literature review (LR) is very limited and statements like, "few studies distinguish between selfies taken and selfies shared..." would need proper referencing. Also, some statements like, "It is clear that marginalized individuals use selfies to achieve a sense of control in an environment wherein control is systematically denied." looks very detached in absence of any explicit enquiry and reference(s). Did the author(s) or anyone checked for it?

   Again, the author(s) wants to "sharing taking and share selfies across different scenarios and consider different types of selfiers,", which is fine but the question is why? please present a strong argument to support your study with clear hypotheses around psychological well-being (greater self-esteem, less reliance on external 140 feedback, greater internal sense of control and mediating control variable). Each construct would need some matter to establish its relevance based on the theory in this study context.

     The methodology is fine and well thought...great work !

     I would also suggest having a research framework that Cleary shows the relationship between the constructs including mediation. Also, including a structural model would be a good idea. 

      I hope that my comments will help the author (s) to improve this work (which is very interesting).

      Best Wishes, 

      Regards,

Author Response

Thank you to the Editor and Reviewers at Behavioral Science for their consideration of our manuscript, “Discerning Selfiers: Differences between taking and sharing selfies” (behavsci-3082030). We have carefully reviewed the comments and have edited our manuscript accordingly. We believe that the current version is a marked improvement and we hope that we have addressed the reviewers’ concerns. 

Below, we have outlined our responses. The reviewer’s comments are in bold, our responses are in regular print, and insertions into the article are in italics and indented. 

Comment 1 I thank the Editor for giving me this opportunity to go through this manuscript. I also applaud the author(s) for the efforts.

Response 1 Thank you!

Comment 2 The introduction of the paper is good but there lacks some clarity as to why it is important to conduct this research. Strong enough arguments are needed to justify the need of the research/study. Please note that without a clear introduction, your readers will struggle. They may feel confused when they start reading your paper.

A good introduction should have Scope: The topic you’ll be covering; Context: The background of your topic and Importance: Why your research matters in the context of an industry or the world, to guide your reader from a general subject area to the narrow topic that your paper covers. Please neatly address the valid gap(s) and research objectives of the paper based on the strong motivation. Include research questions that address the gap(s) identified.

Response 2 Thank you for your detailed instructions regarding the layout of the introduction. We have revamped the introduction significantly and moved some of the claims buried in the literature review to earlier in the manuscript and moved the detailed literature findings into subsections of the introduction. Specifically, in the first paragraph we have addressed…

The scope and context (i.e., selfie motivations and effects of different types of behaviors)

People who take selfies are motivated by self-approval and belonging [1,2], impression management [3], attention seeking, communication, and entertainment [4], as well as documentation or archiving [1,4,5]. Although several studies explore selection and editing behaviors [6,7], only a few studies distinguish between taking and sharing selfies [8–10] and sharing selfies is considered to be inherent to or definitive of selfie use.

And the importance (why our research matters in real world context)

Disregarding the importance of deliberation in making personal moments available to a mass audience limits our understanding of the unique processes, drives, and relationships underlying self-documentation and self-representation via selfies especially given that selfies are powerful opportunities for self-reflection, indicating both an opportunity for “mass communication and interpersonal communication simultaneously” [11].  

We then go on to explicate our research questions in the second paragraph of the introduction:

This research connects two important bodies of work: qualitative or content analysis work assessing different selfie genres, and traditional quantitative psychological surveys investigating motivation and individual differences. Using a novel approach that assesses differences in taking and sharing across different scenarios, the current study poses the following questions: (1) Is taking selfies (i.e., self-documentation) a distinctly different behavior from sharing selfies (i.e., self-representation)? And (2) do people who share selfies deliberately exhibit different patterns between constructs?

Comment 3 Please note that theoretical grounding and the literature review (LR) is very limited and statements like, "few studies distinguish between selfies taken and selfies shared..." would need proper referencing.

Response 3 Thank you for this reminder. We added these citations accordingly throughout the document and incorporated a 2024 meta analysis that also helps explicate trends in the prior literature.

Felig RN, Goldenberg JL. Selfie-evaluation: A meta-analysis of the relationship between selfie behaviors and self-evaluations. Pers Soc Psychol Bull. 2024;50(8):1227–50.

Given that this is incorporated throughout the manuscript, we have not included a specific grab from the text, but please check out our list of citations that has been systematically reorganized to address this concern and highlight studies that both amplify the handful of studies that have taken this approach.

Comment 4 Also, some statements like, "It is clear that marginalized individuals use selfies to achieve a sense of control in an environment wherein control is systematically denied." looks very detached in absence of any explicit enquiry and reference(s). Did the author(s) or anyone checked for it?

Response 4 Thank you for this important note. We have reorganized the literature review so that this line is explicitly summarizing some of the findings described with respect to marginalized selfiers. In the earlier iteration, it was placed in the paragraph after the associated literature.

From Section 1.2 Paragraph 2 (Personal and Masspersonal Photography)

Women post more selfies than men [20, 48, 49], adolescents and younger adults post more selfies than older adults [19], [49], and non-exclusively heterosexual people post more selfies than heterosexual people [19]. Similarly, qualitative research showcases how marginalized racial and ethnic groups use selfies to share their own stories and experiences without the gatekeeping of traditional media institutions [50–53]. Selfies are a form of identity affirmation [54]; through this lens,  discriminated groups, including women, younger people, and LGBTQ+ individuals, use selfies to affirm and amplify their own representation [53, 55–58]. Marginalized individuals use selfies as a means of attaining control through digital technologies to combat systematic denial of control in society [59, 60] and understanding the patterns in taking and sharing of selfies can help illuminate how this practice connects to reclaiming a sense of control. 

Comment 5 Again, the author(s) wants to "sharing taking and share selfies across different scenarios and consider different types of selfiers,", which is fine but the question is why? Please present a strong argument to support your study with clear hypotheses around psychological well-being (greater self-esteem, less reliance on external feedback, greater internal sense of control and mediating control variable).

Response 5 Thank you for this clarification request. We also acknowledge that there was a typo in the original manuscript that you copied. The second paragraph of our introduction states our research questions and the final paragraph of our introduction connects the gaps in the prior literature and explicates our hypotheses. 

From Section 1 Paragraph 2

Using a novel approach that assesses differences in taking and sharing across different scenarios, the current study poses the following questions: (1) Is taking selfies (i.e., self-documentation) a distinctly different behavior from sharing selfies (i.e., self-representation)? And (2) do people who share selfies deliberately exhibit different patterns between constructs?

From Section 1.2 Paragraphs 3-4

…we posit that taking and sharing are two related but distinct behaviors (H1);  (i.e., whereas taking is motivated by self-documentation, sharing is motivated by self-representation [2]) and different patterns of taking and sharing will emerge in different types of selfiers (H2). 

Further, we explore the correlation between selfie behaviors and control by selfier type to understand agentic benefits of the selfie-making process. Given the prior research regarding deliberative selfie making and selfie practices among marginalized populations, we hypothesize that selfiers who report greater likelihood to take than share selfies will report more positive psychological well-being (e.g., less reliance on external feedback, greater self-esteem, H3) through greater reported control (H3a).  

Comment 6 The methodology is fine and well thought...great work !

Response 6 Thank you for these words of encouragement!

Comment 7 I would also suggest having a research framework that clearly shows the relationship between the constructs including mediation. Also, including a structural model would be a good idea. 

Response 7 Thank you for this suggestion. It is important to note that our framework focuses on different types of selfiers - or groups of people who selfie with similar patterns of behavior - not specific types of behaviors, therefore the structural equation model is rooted in a comparison of two different selfie types (i.e., discerning vs. non-discerning). We have added a figure to help clarify. 

Comment 8 I hope that my comments will help the author (s) to improve this work (which is very interesting).

Response 8 Thank you for the positive and valuable feedback!

Round 2

Reviewer 3 Report

Comments and Suggestions for Authors

I am satisfied with the revision!

Best wishes !

Author Response

Thank you for this!